# Characterizing the hygroscopicty of growing particles in the Canadian Arctic summer

Rachel Y.-W. Chang[1], Jonathan P.D. Abbatt[2], Matthew C. Boyer[1,3], Jai Prakash Chaubey[1], and Douglas B. Collins[2,4]

[1]Department of Physics and Atmospheric Science, Dalhousie University, Halifax, Nova Scotia B3H 4R2, Canada
[2]Department of Chemistry, University of Toronto, Toronto, Ontario M5T 1T6, Canada
[3]Now at Institute for Atmospheric and Earth System Research, INAR/Physics, University of Helsinki, Finland
[4]Now at Department of Chemistry, Bucknell University, Lewisburg, Pennsylvania 17837, USA

**Correspondence:** Rachel Y.-W. Chang (rachel.chang@dal.ca)

**Abstract.** The impact of aerosols on clouds is a well-studied, although still poorly constrained, part of the atmospheric system. New particle formation (NPF) is thought to contribute 40–80% of the global cloud droplet number concentration, although it is extremely difficult to observe an air mass from NPF to cloud formation. NPF and growth occurs frequently in the Canadian Arctic summer atmosphere, although only a few studies have characterized the source and properties of these aerosols. This study presents cloud condensation nuclei (CCN) concentrations measured on board the *CCGS Amundsen* in the eastern Canadian Arctic Archipelago from 23 July to 23 August 2016 as part of the Network on Climate and Aerosols: Addressing Uncertainties in Remote Canadian Environments (NETCARE). The study was dominated by frequent ultrafine particle and/or growth events, and particles smaller than 100 nm dominated the size distribution for 92% of the study period. Using $\kappa$-Köhler theory and aerosol size distributions, the mean hygroscopicity parameter ($\kappa$) calculated for the entire study was 0.12 (0.06–0.12, 25th–75th percentile), suggesting that the condensable vapours that led to particle growth were primarily slightly hygroscopic, which we infer to be organic. Based on past measurement and modelling studies from NETCARE and the Canadian Arctic, it seems likely that the source of these slightly hygroscopic, organic, vapours is the ocean. Examining specific growth events suggests that the mode diameter ($D_{max}$) had to exceed 40 nm before CCN concentrations at 0.99% SS started to increase, although a statistical analysis shows that CCN concentrations increased 13–274 cm$^{-3}$ during all ultrafine particle and/or growth times (total particle concentrations > 500 cm$^{-3}$, $D_{max}$ < 100 nm) compared to Background times (total concentrations < 500 cm$^{-3}$) at SS of 0.26–0.99%. This value increased to 25–425 cm$^{-3}$ if the growth times were limited to times when $D_{max}$ was also larger than 40 nm. These results support past results from NETCARE by showing that the frequently observed ultrafine particle and growth events are dominated by a slightly hygroscopic fraction, which we interpret to be organic vapours originating from the ocean, and that these growing particles can increase the background CCN concentrations at SS as low as 0.26%, thus pointing to their potential contribution to cloud properties and thus climate through the radiation balance.

# 1 Introduction

The Arctic environment is changing at a rapid pace, driven by surface air temperatures that are increasing up to two or three times faster than the global average (Serreze and Barry, 2011) and summer sea ice extent that is steadily declining (Stroeve and Notz, 2018; Brennan et al., 2020). Many of these changes are driven by factors that affect the summertime surface energy budget, when solar radiation is present and changes in surface albedo due to melting surface ice or snow can affect the balance in the surface energy. In addition, clouds and aerosols can influence the amount of incoming solar radiation that reaches the surface of the earth. The role of low-level clouds in the summer Arctic radiation budget is complex, as they can cause the surface to not only cool, as expected at lower latitudes, but also warm, depending on the surface albedo and solar zenith angle (Shupe and Intrieri, 2004). Liquid cloud droplets form on pre-existing aerosol particles suspended in the atmosphere. Particles that can activate into droplets are called cloud condensation nuclei (CCN), and whether a given particle will act as a CCN depends on its size, chemical composition and the supersaturation (SS) to which it is exposed. Therefore, it is important to characterize Arctic aerosol properties and their role in cloud formation processes to better understand the surface radiation budget in the changing Arctic atmosphere.

Aerosol concentrations in the Arctic experience a strong seasonal cycle, with long range transport from southerly latitudes bringing high mass concentrations during the winter and spring time, which is known as Arctic Haze. In contrast, the summer Arctic atmosphere is much cleaner with very low aerosol concentrations, providing conditions favourable for new particle formation (NPF). In the marine atmosphere at lower latitudes, NPF has been observed to occur in the free troposphere, but over open waters in the Arctic summer, frequent drizzle reduces the aerosol concentrations (condensation sink) to favour NPF in the boundary layer (Croft et al., 2016a; Browse et al., 2012). Modelling studies have estimated that NPF can contribute 40–80% of cloud droplets globally (Merikanto et al., 2009; Kuang et al., 2009; Pierce and Adams, 2006). However, most of those studies assume that all particles that reach a minimum size, typically 70–100 nm, will automatically contribute to cloud droplet number concentration (CDNC). In determining the contribution of NPF on cloud droplet formation, it is important to characterize: 1) the source of the condensing vapours that lead to NPF and subsequent growth, noting that these sources could be different; 2) the chemical composition, and therefore the hygroscopicity, of the growing particles to determine their ability to activate as CCN; 3) whether the particles grow large enough to activate at a SS that is relevant in the ambient atmosphere; and 4) whether the particles formed are exposed to a SS that allow them to activate. It is extremely rare to actually observe an entire event from NPF to cloud droplet formation since the temporal and spatial scales usually cannot be captured by a single measurement platform and most studies, including this one, can only contribute to investigating some of these aspects.

To understand the sources and chemical composition of summer Arctic aerosols, early studies at Alert, Canada identified large mass fractions from methane sulphonate (MSA) and sulphate, suggesting that secondary aerosol mass originated from the marine biological production of dimethyl sulphide (DMS) and its subsequent atmospheric oxidation products led to aerosol growth (Li et al., 1993). Observations from 2011–2012 at Alert (Leaitch et al., 2013) and during the International Polar Years (2007–2008) in the Canadian Arctic Archipelago (Rempillo et al., 2011; Chang et al., 2011) linked small and growing particles to sulphate and DMS through measurements and modelling, further supporting the role of marine biology in Arctic

aerosol through DMS oxidation in NPF and growth. There is evidence that iodine (e.g. iodine oxoacids) can also contribute to nucleation and growth on the coast of Greenland (Allan et al., 2015; Dall'Osto et al., 2018; Sipilä et al., 2016), as well as more recently in the Central Arctic Ocean (Baccarini et al., 2020) and under controlled laboratory settings (He et al., 2021). However, the observed iodine is not always sufficient to explain the observed growth (Baccarini et al., 2020). Similarly, $NH_3$ emitted from the guano of migratory birds is thought to significantly contribute to NPF in the Canadian Arctic (Croft et al., 2016b; Wentworth et al., 2016).

More recently, a growing body of work is providing evidence that organic compounds contribute to the growth of newly formed particle at lower latitudes (Ehn et al., 2014; Sihto et al., 2011), as well as in the Arctic. In the Canadian Arctic, direct measurements of chemical composition using aerosol mass spectrometry from aircraft (Willis et al., 2016) and at a ground site (Tremblay et al., 2019) showed significant organic contributions during three observed growth events. These growing particles have been linked to open waters at the local to regional scale through footprint sensitivity analyses, suggesting that the source of these organics was from the ocean. Indeed, oxygenated volatile organic compounds (OVOCs) have been observed in the Canadian Arctic (Mungall et al., 2017), with their atmospheric concentrations correlated to sunlight and surface ocean coloured dissolved organic matter. These OVOCs would be less volatile and could, with short amounts of atmospheric ageing, potentially condense and contribute to aerosol growth if the conditions were favourable (Ehn et al., 2014). On the Fram strait, hygroscopicity measurements at super- and sub-saturated conditions have consistently shown that summer aerosols, which are dominated by NPF and growth (Tunved et al., 2013), are less hygroscopic than pure inorganics, with observed hygroscopicity parameters ($\kappa$) ranging from 0.15–0.4 (Zábori et al., 2015; Kecorius et al., 2019; Lange et al., 2019). Since the expected $\kappa$ values for ammonium sulphate and sulphuric acid range from 0.6–0.7 (Petters and Kreidenweis, 2007; Shantz et al., 2008), the observed $\kappa$ values suggest that the aerosol was composed of a significant organic fraction. Similarly, aerosol hygroscopicity measured in the Pacific portion of the Arctic Ocean were 0.08–0.15 (Park et al., 2020; Herenz et al., 2018). A recent modelling study incorporating many of these results found that a large source of organic vapours originating from Arctic waters could explain much of the growth observed after NPF in the Canadian Arctic summer (Croft et al., 2019).

An aspect to consider is whether particles formed from NPF also grow to sizes large enough to activate at atmospherically-relevant SS (0.2-0.6% in the Arctic (Leaitch et al., 2016; Bulatovic et al., 2021)). Cloud studies often find CDNC correlating with the accumulation mode (e.g. Hegg et al., 2012; Jia et al., 2019). However, it has been hypothesized that the extremely low aerosol number concentrations sometimes found in the summer Arctic causes even the smallest particles to quickly activate in a CCN-limited regime (Mauritsen et al., 2011). In-cloud observations from aircraft in the Canadian Arctic (Leaitch et al., 2016) and a mountain site on Svalbard (Koike et al., 2019) have reported that particles as small as 30 nm could be activating when the concentrations were less than 30 $cm^{-3}$, suggesting that newly formed particles need only to grow to 30 nm before contributing to CDNC and potentially affecting climate. Although not direct measurements of cloud droplets, measured CCN concentrations have been observed to increase at SS > 0.1% during NPF and growth in Arctic (Willis et al., 2016; Kecorius et al., 2019; Lange et al., 2019) and Antarctic (Kim et al., 2019) locations, further suggesting that these newly formed particles can activate and contribute to climate effects. In addition, recent modelling work has shown that the SS in high Arctic clouds

can reach as high as 1% if aerosol concentrations are low, further supporting the potential climatic importance of newly formed
particles (Bulatovic et al., 2021).

Although many of the above-mentioned studies have characterized the hygroscopicity of the summer Arctic aerosol, only
four have focussed specifically on studying NPF and growth (Willis et al., 2016; Burkart et al., 2017; Kecorius et al., 2019;
Lange et al., 2019), with the first three studies reporting results from five individual events. In this study we expand these
findings to more events by presenting aerosol and CCN measurements sampled from the *CCGS Amundsen* in the Canadian
Arctic Archipelago during the summer of 2016 as a part of the Network on Climate and Aerosols: Addressing Uncertainties in
Remote Canadian Environments (NETCARE). While previous work by Collins et al. (2017) characterized the ultrafine particle
(UFP) and growth events observed during the cruise, this study focuses on understanding the hygroscopicity of the particles
during growth to determine if the particles grew sufficiently large to activate at a SS that is relevant for cloud formation.
Calculating the hygroscopicity parameter also allows us to characterize the chemical composition of the aerosols and thus infer
the sources of the condensing vapours that contributed to the growth of the UFP.

## 2   Methods

### 2.1   Sampling platform and instrumentation

This study uses data collected from the NETCARE ship-based observations on board *CCGS Amundsen* from 14 Jul to 25
Aug 2016, with the cruise track shown in Fig. 1. Based on the proximity to the open ocean, surrounding land mass, time, and
direction of the ship cruise, the study was conveniently divided into three legs based on geography: (i) Baffin Bay (23 Jul – 4
Aug), (ii) Nares Strait (5–17 Aug), and (iii) Resolute Bay (17–23 Aug), shown in Fig. 1 as the red, blue, and green coloured
boxes, respectively. While the ship was in the Baffin Bay region, it was influenced more by the surrounding open waters. In
contrast, the ship was closer to sea-ice or land masses in Legs 2 and 3, respectively. Collins et al. (2017) showed that the
5-day backward simulations of surface ($< 200$ m) influence using the FLEXible PARTicle dispersion model (FLEXPART) was
primarily confined to the immediate region of the ship, suggesting that regional emissions were important during our study.
However, it is difficult to separate the effects of geography and season on the observed differences in the aerosol population
since the local conditions in each region (e.g. sea-ice coverage, solar radiation, biological activity, cloud cover, wind speed)
would have affected the observed aerosols.

Ambient measurements of CCN concentrations were made with a streamwise thermal gradient CCN counter (CCNC,
Droplet Measurement Technologies, CCN-100) (Roberts and Nenes, 2005). Ambient aerosols measured by the CCNC were
sampled from an inlet approximately 38 m from the bow on the port side, approximately 9 m above sea level. The CCNC was
operated at a total flow rate of 0.5 L min$^{-1}$ and a sample flow rate of 0.05 L min$^{-1}$ with an estimated inlet residence time of
1.5 s and particle loss rate of $< 2\%$ for the particles that activated in the chamber ($>40$ nm) (von der Weiden et al., 2009).
The temperature gradient of the CCNC was varied such that ambient aerosols were exposed to SS of 0.17, 0.26, 0.44, 0.63,
0.81 and 0.99%, where each SS was maintained for 10 min at a time. The SS of the chamber were calibrated weekly during
the observation period with ammonium sulphate aerosols at 5 SS values between 0.1–1%, except on 18 August 2016 when it

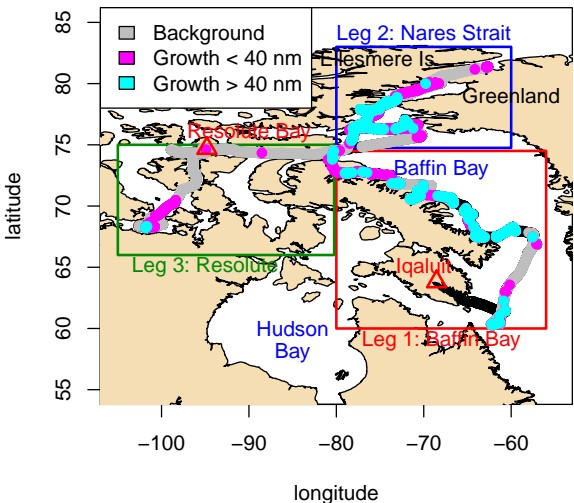

**Figure 1.** Track of the NETCARE ship cruise (black and coloured points). Rectangles show the three legs and red triangles denote communities. The colour of the points show times used in calculations of Background (grey) and Growth > 40 nm (light blue). Additional times included in the UFP and Growth calculations are shown in magenta.

was only calibrated at 0.44 and 0.99% SS. The SS was stable throughout the 4 weeks of observations, although the plateau of the calibration curves decreased throughout the study. This could be interpreted as a reduction in the counting efficiency of the CCNC, although after extensive analysis, we believe it is more likely due to issues with the calibration system. As such, we
assume that the ambient CCN concentrations were correct, and the analysis in this study used the values directly reported by the CCNC without accounting for any additional particle loss or counting efficiency corrections.

A scanning mobility particle sizer (SMPS, TSI Inc., Model 3080, 3787) measured the number size distributions of aerosol between the diameter range of 10 and 430 nm, with a time resolution of 5 min, and sample and sheath flow rates of 0.6 and 6 L min$^{-1}$, respectively. This instrument was in the foredeck container, approximately 31 m from the CCNC. Further details
on the SMPS sampling methods, data accuracy and data filtering techniques for ship pollution are presented by Collins et al. (2017). The results presented in this study use observations from 23 July to 23 August 2016 when measurements from both instruments were available.

## 2.2   CCN data analysis

The first 30 s of CCN data at every SS were excluded because the chamber temperatures require time to stabilize when changing
to a new SS. Only values with the temperature stabilized flag, as reported by the CCNC, were used in this analysis. Additionally, the sample times that were previously identified to be contaminated by Collins et al. (2017) were excluded from the CCNC data.

It has been reported that the total uncertainty in the estimated CCN concentration above $100 \, \mathrm{cm}^{-3}$ varies between 7 and 16% due to factors such as temperature, pressure, flow, etc. (Moore et al., 2011). One second CCN concentrations were matched to the 5 min SMPS sample times and the median concentrations calculated. The median was used to avoid the influence of outliers that may have remained after filtering and are, on average, only 1.2% lower than mean concentrations. Of the 2061 concurrent SMPS and CCN observations, 69 were excluded due to high variability (standard deviation to median ratio was greater than 0.5) or the concentrations exceeded $2000 \, \mathrm{cm}^{-3}$.

## 2.3  $\kappa$ calculations

The hygroscopicity parameter ($\kappa$) was calculated using the aerosol size distribution, CCN concentrations and $\kappa$-Köhler theory (Petters and Kreidenweis, 2007):

$$\kappa = \frac{4A^3}{27D_a^3(ln(S_c))^2}, \tag{1}$$

where $A = (4M_w)/(RT\rho_w)$, $M_w$ and $\rho_w$ are the molar mass and density of water, respectively, while $R$ and $T$ are the ideal gas constant and the absolute temperature, respectively. $D_a$ is the activation diameter and represents the diameter above which particles are large enough to activate as CCN at a saturation ratio of $S_c$. $D_a$ was determined by integrating aerosol size distributions downwards from the largest size bin until the cumulative particle number concentration was equal to the corresponding CCN concentration. The size bin at which this occurred was the $D_a$. This method has been previously used in studies where it is not possible to size select the particles before they are counted by the CCNC (e.g. Collins et al., 2013; Gao et al., 2020). Critical assumptions in this approach are that the particles are uniform in composition at a given size and that all the particles larger than $D_a$ activate. These assumptions are unlikely to significantly affect our calculations because the summer Arctic aerosol typically originates from local sources, reducing the likelihood of a separate mode with different properties transported from afar. In addition, although primary emissions from Arctic sources, such as sea spray or dust, could potentially contribute to the aerosol being externally-mixed, these particles would be expected to be present at larger sizes and not affect the calculation of $D_a$.

It is possible that the more volatile fractions of the ambient aerosol could have evaporated in the inlet line or in the CCNC. The room temperature was approximately 4–6 K warmer than the outside air temperature, the top of the CCNC's chamber was approximately 2–3 K warmer than the inlet temperature, and the bottom of the chamber was up to 20 K warmer than the inlet temperature, although a significant amount of aerosol liquid water would have been present by that point. The effect of a 20 K warming has been observed to reduce the CCN-activity of laboratory-generated secondary organic aerosol (SOA), suggesting that the volatile fraction can be more hygroscopic (Asa-Awuku et al., 2009). On the other hand, the most volatile fraction of ambient SOA has been observed to be less oxidized (Huffman et al., 2009), which we would expect to be less hygroscopic (Massoli et al., 2010; Lambe et al., 2011; Chang et al., 2010). More recent results of CCN measurements behind a thermal denuder inferred that the $\kappa$ value for ambient organics at an urban site increased with heating of 20 K (Mei et al., 2021) suggesting that the volatile fraction was less hygroscopic. Overall, it seems that the effect of heating on aerosol hygroscopicity depends on the specific nature of the organic aerosol present and it is unclear whether the hygroscopicity estimated in our

study are potentially over- or under-estimated. As such, no corrections were applied for this warming, although it should be
       considered when interpreting our results.

## 2.4  Defining growth and background times

       To determine the overall contribution of growing UFP on the CCN concentrations throughout this study, UFP and/or growth
       times were defined as times when the total number concentrations from the SMPS ($N_T$) > 500 cm$^{-3}$ and $D_{max} \leq 100$ nm
(magenta and light blue points in Fig. 1), where $D_{max}$ is the statistical mode diameter in the size distribution (i.e. the diameter
       that had the maximum normalized concentration). An additional period when $N_T > 500$ cm$^{-3}$ and $40 nm < D_{max} < 100 nm$
       was defined as Growth > 40 nm (light blue points in Fig. 1), to examine CCN concentrations when the particles had grown larger
       than 40 nm but remained smaller than 100 nm. Both of these periods were compared to periods when minimal contributions
       from UFP and growth were expected. These Background periods were defined as times when $N_T \leq 500$ cm$^{-3}$ (grey points
in Fig. 1). To retain clarity, periods when $N_T > 500$ cm$^{-3}$ and $D_{max} \leq 40$ nm (magenta points in Fig. 1) will be referred to
       as Growth < 40 nm in the legends, although these events do not always show growth and are not considered separately. The
       concentration limit of 500 cm$^{-3}$ was determined by considering various statistical values including the median $N_T$ over the
       entire study (472 cm$^{-3}$) and the 95[th] percentile of $N_T$ when $D_{max} > 100$ nm (484 cm$^{-3}$).

## 3  Results and discussion

## 3.1  General overview

       The aerosol size distributions measured by the SMPS throughout the study and $D_{max}$ are shown in Fig. 2a, and the CCN
       concentrations and $N_T$ are shown in Fig. 2b. The three legs are denoted by the shaded boxes above panel a. Summary statistics
       of CCN concentrations for the entire study and each leg are shown in Table 1. As seen in Fig. 2a, the study was characterized
       by frequent UFP and growth events, which caused $N_T$ to vary by three orders of magnitude. These events were especially
prevalent in the first part of the study when the ship was in the warmer and more biologically-active waters of Baffin Bay
       and persisted into the southern portion of Nares Strait into Kane Basin of the 2nd leg. The most northern extent of the cruise
       occurred on 13 August, where the ship encountered sea-ice and very few UFP were observed. Only one UFP and/or growth
       event was observed during the last leg of the study where the waters were ice-free, shallower and less saline. It is possible that
       the local conditions were less favourable for the formation of UFP, although it should be noted that this period was much shorter
than the other legs (six days compared to 12–13 days). Collins et al. (2017) identified 14 UFP and growth events during this
       cruise which accounted for 41% of the sample times. However, particles smaller than 100 nm dominated the SMPS number size
       distribution for 92% of the study, suggesting that most of the aerosol particles observed throughout the study had undergone
       secondary formation processes such as condensational growth. As such, even if particles were not actively growing at a given
       time, their CCN-activity and inferred chemical composition could still provide insight into the vapours that contributed to
particle growth. In addition, any externally-mixed aerosol that existed before these growth events would become more similar

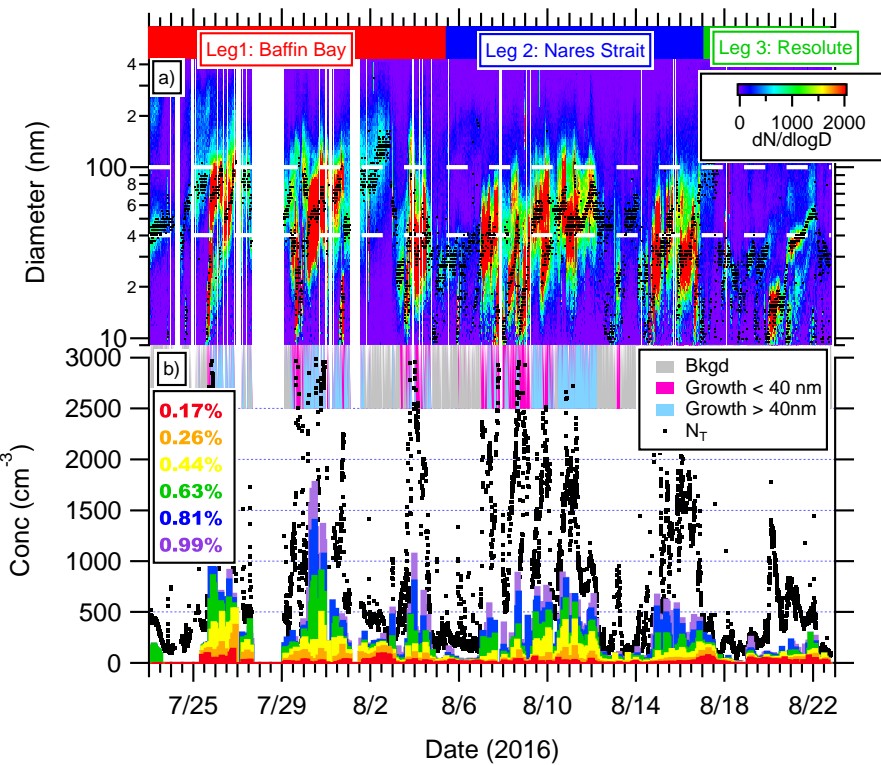

**Figure 2.** Time series of a) aerosol number size distribution and $D_{max}$ (black points) and b) $N_T$ (black points) and CCN concentrations. Dashed white horizontal lines show particle diameters of 40 and 100 nm. Times used in calculations of Background, Growth > 40 nm and Growth < 40 nm are shown in grey, light blue, and magenta, respectively.

in composition due to condensational growth, thus reducing errors associated with the assumptions used to calculate $\kappa$ using Eq. 1.

CCN concentrations at the higher SS usually varied according to $N_T$, showing evidence that UFP and growth can lead to increased CCN concentrations. However, the median activation ratio (AR), defined as the CCN concentration / $N_T$, was only 205  0.38 at the highest SS of 0.99%, suggesting that most of the particles larger than 10 nm were either too small and/or non-hygroscopic to activate. These low AR are consistent with observations from Ny Alesund (Jung et al., 2018), where summer time AR of 0.4 for particles between 10 and 561 nm were reported, as well as other Arctic sites with a similar lower cut-off for $N_T$ (Table 2). Overall, these low AR values reflect the prevalence of particles smaller than 40 nm, especially in the Baffin Bay and Nares Strait regions. The AR increased during the third leg where the waters were ice-free, shallower and less saline when 210  only one UFP and/or growth event occurred.

Over the entire study, the median and interquartile range of CCN concentrations increased with increasing SS (see Table 1), with a median CCN concentration of 29 cm$^{-3}$ at SS of 0.17% compared to 98 and 228 cm$^{-3}$ at 0.44% and 0.99%, respectively. To put our observations into a global perspective, CCN and $N_T$ concentrations observed at a select number of polar and remote

**Table 1.** Summary of CCN concentrations and associated calculations, given as median (25th-75th percentile)

| SS | 0.17% | 0.26% | 0.44% | 0.63% | 0.81% | 0.99% |
|---|---|---|---|---|---|---|
| | | | CCN [cm$^{-3}$] | | | |
| All | 29 (20-45) | 54 (35-92) | 98 (48-171) | 141 (80-211) | 197 (89-353) | 228 (99-456) |
| Baffin Bay | 36 (20-57) | 92 (48-124) | 162 (98-234) | 158 (127-221) | 244 (153-441) | 286 (184-549) |
| Nares Strait | 22 (16-29) | 40 (27-59) | 89 (38-156) | 140 (61-264) | 209 (85-386) | 280 (79-496) |
| Resolute Bay | 41 (32-53) | 56 (49-68) | 70 (49-92) | 87 (67-120) | 118 (78-194) | 136 (89-231) |
| | | | AR | | | |
| All | 0.05 (0.02-0.13) | 0.08 (0.04-0.18) | 0.18 (0.09-0.27) | 0.31 (0.16-0.39) | 0.32 (0.2-0.48) | 0.38 (0.24-0.52) |
| Baffin Bay | 0.05 (0.02-0.09) | 0.09 (0.05-0.2) | 0.20 (0.1-0.33) | 0.36 (0.24-0.4) | 0.37 (0.21-0.5) | 0.40 (0.3-0.48) |
| Nares Strait | 0.02 (0.01-0.07) | 0.05 (0.03-0.1) | 0.13 (0.08-0.22) | 0.21 (0.13-0.31) | 0.27 (0.18-0.37) | 0.34 (0.21-0.46) |
| Resolute Bay | 0.16 (0.07-0.22) | 0.24 (0.12-0.36) | 0.27 (0.15-0.43) | 0.33 (0.17-0.57) | 0.47 (0.24-0.69) | 0.54 (0.32-0.86) |
| | | | $D_a$ | | | |
| All | 163 (141-175) | 122 (113-131) | 91 (85-102) | 76 (68-102) | 64 (57-76) | 57 (50-71) |
| Baffin Bay | 178 (162-195) | 131 (122-141) | 98 (88-113) | 91 (76-118) | 74 (64-91) | 71 (57-82) |
| Nares Strait | 157 (141-168) | 122 (118-131) | 88 (82-95) | 71 (66-79) | 62 (56-68) | 55 (50-64) |
| Resolute Bay | 131 (114-157) | 113 (102-122) | 95 (79-102) | 71 (56-84) | 62 (43-74) | 50 (41-62) |
| | | | $\kappa$ | | | |
| All | 0.11 (0.09-0.17) | 0.11 (0.09-0.14) | 0.09 (0.06-0.11) | 0.08 (0.03-0.11) | 0.08 (0.05-0.11) | 0.07 (0.04-0.11) |
| Baffin Bay | 0.08 (0.06-0.11) | 0.09 (0.07-0.11) | 0.07 (0.05-0.1) | 0.05 (0.02-0.08) | 0.05 (0.03-0.08) | 0.04 (0.03-0.07) |
| Nares Strait | 0.12 (0.10-0.17) | 0.11 (0.09-0.12) | 0.10 (0.08-0.12) | 0.10 (0.07-0.12) | 0.09 (0.06-0.12) | 0.08 (0.05-0.11) |
| Resolute Bay | 0.21 (0.12-0.31) | 0.14 (0.11-0.19) | 0.08 (0.06-0.14) | 0.10 (0.06-0.19) | 0.09 (0.05-0.26) | 0.11 (0.06-0.20) |

sites are shown in Table 2. This list is by no means exhaustive, but shows that overall, our CCN concentrations are generally
higher than other Arctic observations at Ny Alesund (Jung et al., 2018) and the Central Arctic Ocean (Martin et al., 2011).
A notable exception is the aircraft observations over the Canadian Arctic Archipelago in 2008 which were influenced by
biomass burning and transport of industrial emissions and resulted in higher aerosol concentrations (Lathem et al., 2013).
Our observations are closer to those reported for a sub-Arctic Canadian site in the spring (Herenz et al., 2018) and Antarctica
(Herenz et al., 2019; Yu and Luo, 2010). In addition, our observations are lower than measurements from high altitude locations
at lower latitudes (Gogoi et al., 2015; Jurányi et al., 2010).

Calculated $\kappa$ values and summary statistics are also presented in Table 1. Overall, the values were very low, with medians
for the entire study ranging from 0.07–0.11 over the six SS. Values were highest at the lowest SS (0.08–0.21), suggesting that
the larger particles were more hygroscopic, which is consistent with more processing in the larger aerosols or a different source
resulting in a higher hygroscopicity, such as sea spray. The values also increased throughout the study, with the lowest values
(0.04–0.08) corresponding to the first part of the study when the ship was in the more southerly, warmer and open waters

**Table 2.** CCN and $N_T$ concentrations from other studies

| Season | Platform | CCN (cm$^{-3}$) | SS (%) | $N_T$ | Size range (nm) | Reference |
|--------|----------|-----------------|--------|-------|-----------------|-----------|
| **High Arctic** | | | | | | |
| Summer | Ground | 45–81 | 0.45 | 195 | 10–561 | (Jung et al., 2018) |
| Summer | Ship | 14 ±11 | 0.1 | | | (Martin et al., 2011) |
| | | 47±31 | 0.73 | | | |
| Summer | Aircraft | 247 | 0.5 | 514 | >10 | (Lathem et al., 2013) |
| **Arctic** | | | | | | |
| Spring | Ground | 10–250 | 0.1–0.7 | 20–500 | >10 | (Herenz et al., 2018) |
| **Antarctica** | | | | | | |
| Summer | Ground | 10–1300 | 0.1–0.7 | 40–6700 | >3 | (Herenz et al., 2019) |
| Summer | Ground | 60–200 | 0.4 | ≈2000 | >10 | (Yu and Luo, 2010) |
| **High Altitude Regions** | | | | | | |
| Winter | Nainital | 2180 ±16 | 0.46 | 2891±2020 | >10 | (Gogoi et al., 2015) |
| Spring | Jungfraujoch | 149±171 | 0.12 | 550 | 12–570 | (Jurányi et al., 2010) |
| | | 568±401 | 1.18 | | | |

of Baffin Bay where more UFP and growth events were observed, and the highest values (0.18–0.21) corresponding to the shallower and more coastal waters during Leg 3 when only one UFP and growth event occurred. Based on our observations, it is unclear whether the increase in hygroscopicity driven by a reduction in UFP and growth events was caused by changes in emissions in the different regions such as more land influence in the Resolute leg, or whether it is due to the fact that we sampled in this region two weeks later when the biological activity had changed. It should be noted that the 2014 NETCARE cruise in the Queen Maud Gulf region west of Resolute Bay did not observe any UFP events in early August (Collins et al., 2017) while Chang et al. (2011) reported one UFP and growth event in late August in 2008 in the same region. Overall, this suggests that local environmental conditions are the main driver of aerosol sources and therefore hygroscopicity. The $\kappa$ values calculated from this study are consistent with the NETCARE aircraft observations for one NPF event (0.1) (Willis et al., 2016; Burkart et al., 2017) as well as other mobile platforms over Arctic waters near Svalbard during four NPF and growth events (0.13 for 20 nm particles) (Kecorius et al., 2019) and from air influenced by the Beaufort and Chukchi Seas (0.08) and the Pacific Ocean (0.03) (calculated from Park et al., 2020). However, the observed particles appear less hygroscopic than those observed in the Central Arctic Ocean (0.33±0.11) (Martin et al., 2011), as well as from land-based sites in spring at Tuktoyaktuk, Canada (0.23) (Herenz et al., 2018); non-anthropogenically-influenced particles at the VRS site in northeast Greenland (0.25–0.4) (Lange et al., 2019); and from Zeppelin (0.3–0.46) (Jung et al., 2018; Zábori et al., 2015). The lower values determined from the mobile platforms likely reflect the more recently condensed vapours during individual growth events, whereas the land-based sites generally reported values averaged over several months, causing them to be less sensitive to distinct UFP and growth events.

Based on the median calculated $\kappa$ values from the entire study, we infer that >80% of the aerosol volume fraction was composed of a non-hygroscopic to slightly hygroscopic component, which we interpret as being organic. Previous modelling results have shown that the air sampled during our study was influenced by source regions within the Arctic circle which is mostly marine (Collins et al., 2017; Burkart et al., 2017), suggesting that these slightly hygroscopic aerosols were influenced by the water. Other recent studies in the Canadian Arctic have inferred that VOCs are emitted from the ocean (Mungall et al., 2017), that the source of secondary aerosol mass for UFP and growth are driven by marine biological influences (Collins et al., 2017; Willis et al., 2017), that particles < 100 nm during growth events are almost entirely organic and influenced by open waters (Tremblay et al., 2019; Willis et al., 2016) and that including a significant source of Arctic marine SOA could explain the frequently observed NPF and growth events (Croft et al., 2019). Finally, the reported values are consistent with the $\kappa$ value reported for pure SOA inferred from ambient observations at continental sites (e.g. Sihto et al., 2011; Gunthe et al., 2009). Together, these results suggest that the condensing material contributing to particle growth is slightly hygroscopic, likely organic originating from marine biological sources.

## 3.2  Influence of particle growth on CCN

To further explore the potential role of UFP and growth on climate, CCN concentrations and $D_a$ were examined for specific events when UFP and growth were observed. Figure 3 shows a period when particles smaller than 20 nm appeared on several occasions and then grew to $D_{max}$ of 40–60 nm, with the larger tail of the mode sometimes growing as large as 100 nm (upper dashed white line in panel a). Since the $D_a$ at 0.99% remained between 45 and 70 nm (purple triangles in panel a), CCN concentrations remained low during the two UFP and growth events on 8 Aug, even as total concentrations exceeded 2000 cm$^{-3}$ because the majority of the particles were too small and/or non-hygroscopic to activate. However, as the particle $D_{max}$ grew larger than 40 nm after 9 Aug 08:00 (light blue shading in panel b), CCN concentrations noticeably increased at all SS > 0.44%, suggesting that these growing particles became sufficiently large to activate. The $\kappa$ values determined during this period mostly remained between 0.05 and 0.18 at all SS, suggesting that the hygroscopicity, and therefore chemical composition, was similar across the size range of the $D_a$ determined at all SS (40–180 nm, pink circles and purple triangles in panel a). Similar results are seen in Fig. 4 which shows three UFP and growth events between 29 Jul and 5 Aug. CCN concentrations most obviously increased after 30 July 00:00 when $D_{max}$ exceeded 40 nm (light blue shading), and did not increase when bursts of smaller particles appear at 31 July 18:00 and 03 August 08:00 (magenta shading).

To determine the overall contribution of growing UFP on the CCN concentrations throughout this study, Fig. 5 shows the increase in the mean CCN concentration at each SS during all UFP and/or Growth times (i.e. Growth < 40 nm and Growth > 40 nm) compared to Background times (purple dashed line) as well as the increase during only Growth > 40 nm times compared to Background times (light blue line). The whiskers in Fig. 5 denote the 95% confidence interval calculated as 1.96 times the standard errors of the two means summed in quadrature. During all UFP and/or Growth times, the CCN concentrations increased by 13–161 cm$^{-3}$ at SS of 0.26%–0.99%, corresponding to a 22–167% increase in concentration compared to background, demonstrating that particles formed from UFP and growth events in the Canadian Arctic can contribute to CDNC at a very modest SS of 0.26%. This effect is even more pronounced if only Growth > 40 nm times are considered, when

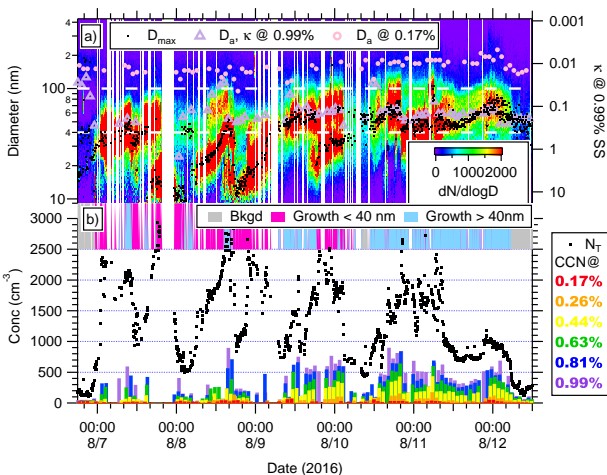

**Figure 3.** As in Fig. 2 but for 07–12 Aug 2016. $D_a$ calculated at 0.17% and 0.99% SS are included in panel a) as well as the corresponding $\kappa$ values calculated for $D_a$ at 0.99% SS.

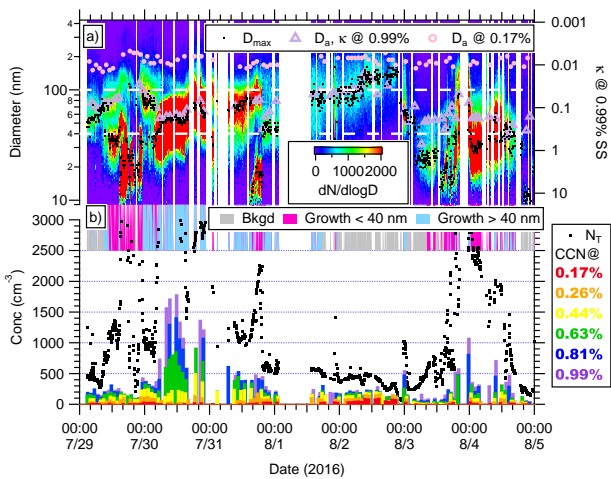

**Figure 4.** As in Fig. 2 but for 29 Jul–05 Aug 2016. $D_a$ calculated at 0.17% and 0.99% SS are included in panel a) as well as the corresponding $\kappa$ values calculated for $D_a$ at 0.99% SS.

CCN concentrations increased by 25–425 cm$^{-3}$ at SS of 0.26%–0.99%, corresponding to a 43–259% increase in concentration compared to Background periods. These values are statistically significant ($p < 0.05$) when tested with a Mann Whitney test (*wilcox.test* function, R version 4.0.3). Because only one UFP event was observed in the last leg of the study when the ship was in the Lancaster Sound and Queen Maud Gulf, these results are only representative of the Baffin Bay and Nares Strait periods of the study. Our findings are consistent with observations on northeast Greenland where CCN concentrations were reported to increase by 42–95 cm$^{-3}$ at SS of 0.3% and 85–150 cm$^{-3}$ at higher SS during the Nascent and Bimodal clusters, which were

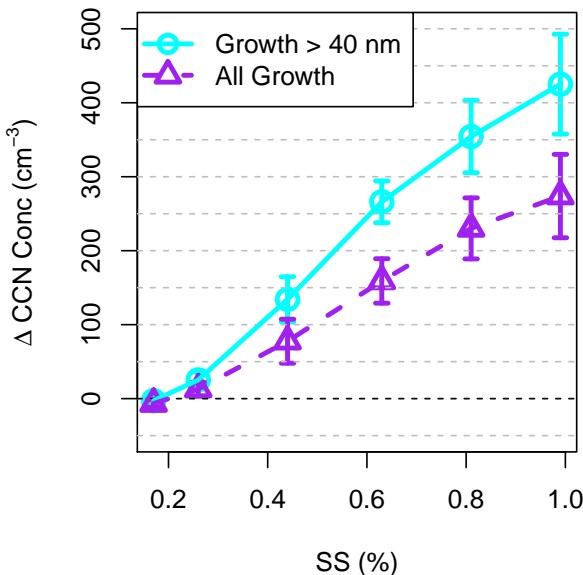

**Figure 5.** The increase in CCN concentration during Growth > 40 nm (cyan) and All Growth (purple) times compared to Background times at different SS. The error bars show the 95% confidence interval.

characterized by NPF (Lange et al., 2019). Our findings also support NETCARE aircraft studies performed during the summer
of 2014 where CCN concentrations increased by $\approx$ 100 cm$^{-3}$ at SS of 0.6% for one growth event (Willis et al., 2016) and particles as small as 30 nm, presumably formed recently, were inferred to have activated at SS of 0.3% in clouds (Leaitch et al., 2016). Similarly, particles as small as 30–50 nm were inferred to activate when aerosol concentrations were less than 30 cm$^{-3}$ at the mountain site Zeppelin on Svalbard (Koike et al., 2019). Together, these studies suggest that small particles, influenced by condensational growth, can affect cloud radiative properties by contributing to CDNC.

## 3.3 Twomey parameterization

Figure 6 shows the mean CCN concentrations at each SS for the full study period (solid black circles) and the three regions considered in the study. The lines in Fig. 6 represent the best fit of the empirical Twomey parameterization often used in models to relate CCN to SS using:

$$CCN = C \times SS^k, \tag{2}$$

where $C$ represents the CCN concentration at 1% SS and $k$ is the power law exponent (Twomey, 1959). The parameters $C$ and $k$ are widely used in cloud microphysical models as they provide information about size and composition of the background aerosol concentration (Seinfeld and Pandis, 1997), although more recent parameterizations have adopted Eq. 2 to include more information about the aerosol microphysical parameters (Khvorostyanov and Curry, 2006). The Twomey parameters calculated

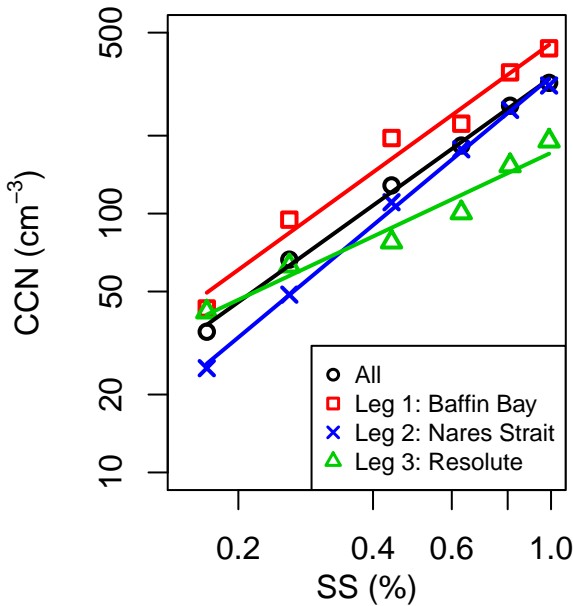

**Figure 6.** Twomey parameterization for the entire study (black) and the three legs (red, blue and green).

**Table 3.** Calculated Twomey parameters

|   | All | Baffin Bay | Nares Strait | Resolute Bay |
|---|-----|------------|--------------|--------------|
| $C$ | 335 | 455 | 337 | 172 |
| $k$ | 1.24 | 1.25 | 1.44 | 0.81 |

for our study are listed in Table 3. The $C$ parameter for the full study period and the Baffin Bay and Nares strait regions are
very similar (between 335 to 455 cm$^{-3}$), whereas it was much lower (172 cm$^{-3}$) for the Resolute Bay region, reflecting the
overall lower $N_T$ during Leg 3. With the exception of the last leg, these values are all significantly higher than the 100–140
cm$^{-3}$ determined for the Zeppelin station at Ny Alesund for July and August (Jung et al., 2018), likely due to the greater
aerosol concentrations caused by UFP and growth during our study resulting in more CCN at 1% SS. Similarly, the maximum
$k$ estimated at Ny Alesund was in July, with a value of 0.5606, which is lower than the values estimated for our study, especially
the Baffin Bay and Nares Strait regions (1.2–1.4). This can also be attributed to the persistent UFP and/or growth events during
our study since a greater number of small particles can activate as CCN at higher SS. In contrast, the Resolute Bay region,
which had less UFP and growth events, had a much lower $k$ of 0.81, showing that the CCN spectrum was less sensitive to
changes in SS.

## 4 Conclusions

This study reports CCN concentrations measured in the eastern portion of the Canadian Arctic Archipelago during the NET-CARE campaign onboard the *CCGS Amundsen*. The observations reported here took place from 23 Jul – 23 Aug 2016 when UFP and/or growth events were highly prevalent, with particles smaller than 100 nm dominating the particle number size distribution for 92% of the study. These UFP and/or growth events resulted in high particle concentrations which were also reflected in increased CCN concentrations, suggesting that the frequently-observed small, growing particles have the ability to contribute to CDNC and therefore the radiative budget in the Arctic. The mode diameter of the growing particles generally had to reach 40 nm before the particles became CCN-active at 0.99% SS, although CCN concentrations were shown to have a statistically-significant increase of 13 and 78 $cm^{-3}$ at even modest SS of 0.26 and 0.44% compared to Background times, defined as when $N_T$ was lower than 500 $cm^{-3}$. The influence of frequent UFP and/or growth events was also reflected in the Twomey parameterization, which resulted in higher $C$ and $k$ values than previous studies.

The calculated $\kappa$ values for the observed CCN provide insight on the hygroscopicity of the particles as well as their composition, and thus the source of the condensing vapour that contributed to the observed aerosol growth. Values were low, with a mean of 0.12 (0.06–0.12, 25–75th percentile) for the entire study over all SS, suggesting that at least 80% of the volume fraction of the aerosol was non-hygroscopic to slightly hygroscopic, likely organic. Since the history of the air mass was generally local or regional, the condensing slightly hygroscopic vapours most likely originated from the open waters within the Arctic. These $\kappa$ values are lower than many other Arctic observations on land, although these other studies reported seasonal means, which likely included more aged aerosols. Our values are more consistent with other observations reported from mobile platforms in open Arctic waters, especially during four UFP and growth events near Svalbard (Kecorius et al., 2019) and one event in the Canadian Arctic (Burkart et al., 2017; Willis et al., 2016), likely reflecting the hygroscopicity of the locally-produced condensing vapours before significant chemical ageing could occur.

This work provides an important link between other publications from the Canadian Arctic, especially from the NETCARE study (Willis et al., 2016, 2017; Burkart et al., 2017; Mungall et al., 2017; Collins et al., 2017), by confirming that the numerous particle growth events observed in summer 2016 was indeed driven by slightly hygroscopic vapours that are presumed to be organic. The present study corroborates prior chemical composition measurements of a small number of events (Willis et al., 2016; Tremblay et al., 2019) and expands their spatial and temporal scope, thus demonstrating that the phenomenon of slightly hygroscopic, likely organic, vapours condensing on UFPs is likely prevalent throughout the region. The presence of an Arctic marine SOA was a fundamental assumption in the work of Croft et al. (2019) when simulating new particle formation and growth in the Canadian Arctic, and our results support this assumption.

Overall, these results are important because it is the only study besides the work of Kecorius et al. (2019), Burkart et al. (2017), and Willis et al. (2016) to characterize the hygroscopicity of specific UFP and growing particles in Arctic waters. Our study demonstrates that the frequently-observed UFP and/or growth in the Canadian Arctic summer increases CCN concentrations by 26% at a modest SS of 0.44%, although further work will be needed to determine the ultimate effect on cloud properties and the radiation budget. The chemical composition, as inferred from the hygroscopicity, of the growing particles

also provides insight on the source of the condensing vapours and allows us to better understand how changing environmental conditions in the Arctic (e.g. increased temperatures, reduced sea ice extent, lower salinities) can alter future aerosol processes, and by extension, the radiation budget.

*Data availability.* NETCARE observation data are available on http://crd-data-donnees-rdc.ec.gc.ca/CCCMA/products/NETCARE/.

*Author contributions.* This study was conceived by RYWC and JPDA; the data were collected by RYWC, DBC and MCB; the data were analyzed by RYWC and JPC; and the manuscript was primarily written by RYWC with contributions from all other authors.

*Competing interests.* The authors declare that they have no competing interests.

*Acknowledgements.* This research was conducted as a part of Network on Climate and Aerosols: Addressing Key Uncertainties in Remote Canadian Environments (NETCARE), funded by the Climate Change and Atmospheric Research (CCAR) program within the Natural Sciences and Engineering Research Council of Canada (NSERC). Additional research funding was provided through an NSERC Discovery Grant (2014-05173) and the Ocean Frontier Institute, through an award from the Canada First Research Excellence Fund. The authors would like to thank T. Papakyriakou as well as Captain A. Gariépy and the crew of *CCGS Amundsen* for their assistance throughout the cruise; the Canadian Foundation for Innovation for funding the instruments; G. Evans, R. Leaitch, S. Sharma and D. Veber for the use of their equipment for calibration; and Environment and Climate Change Canada, K. Levesque and R. Christensen for logistical support throughout the study.

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
