# Peer review of "Characterizing the hygroscopicity of growing particles in the Canadian Arctic summer"

_Atmospheric Chemistry and Physics, 2021_

## Author Comment (AC1)

**Reply to Reviews of "Characterizing the hygroscopicty of growing particles in the Canadian Arctic summer" by Rachel Y.-W. Chang et al.**

Blue text refers to reviewer comments.
Black text are replies to comments.
*Italicized text are excerpts from the manuscripts with changes in red*

We thank the two reviewers for taking the time to read our paper and for providing constructive comments. We have incorporated the reviewers' suggestions into the revised manuscript and provide point-by-point responses to each comment below.

Reviewer 1

This manuscript investigates the CCN activities of artic aerosol, especially the one that originated from NPF and/or with early growth. The measurements were done, during summer 2016, on a vessel, in the Canadian Arctic Archipelago region. CCN is measured using a CCN counter (CCNC Droplet Measurement Technologies, CCN-100). These measurements are part of the NETCARE survey. In this present study, the authors performed a careful study of their data by investigating the potential CCN activities of very specific aerosols. They differentiate two types of aerosols: The one that nucleates and reaches a size below 40 nm, and the one that reaches a size between 40 and 100nm (both with a Ntot bigger than 500 particles per cc). The main idea behind this study is to investigate the effect of NPF and growth on global CCN activities in the artic. The authors found that although these new particles increase slightly the number of CCN compared to the artic aerosol background, their hygroscopicity is rather low, and conclude that the aerosol growth is mainly due to organic material, probably from the sea. The authors provide updated values for the parameters of the Twomey parameterization (CCN parameterization) for aerosol generated in the Canadian archipelago. The paper complements previous studies performed with the NETCARE survey. The manuscript is well structured, concise, and well written. I, therefore, recommend the manuscript for publication in ACP following appropriate response to the following minor comments.

Minor comments/questions:

P2L44: "whether the particles grow large enough to activate at a SS that is relevant in the ambient atmosphere" what it the range of SS relevant in an artic ambient atmosphere?

The study by Leaitch et al. (2016) suggested that low altitude clouds in the Canadian Arctic summer had SS of 0.3% and that high altitude clouds could reach SS of 0.6%. Recent work conducted by Bultovic et al. (2021) using two different large-eddy simulation models for the High Arctic (>80° N) simulated median SS in clouds of 0.2-0.4%, depending on the aerosol size distribution, although they can sometimes reach 1%. The revised manuscript now includes this discussion on lines 79 and 88-90:

"*An aspect to consider is whether particles formed from NPF also grow to sizes large enough to activate at atmospherically-relevant SS (0.2-0.6% in the Arctic (Leaitch et al., 2016; Bultovic et al., 2021))*"

*"In addition, recent modelling work has shown that the SS in high Arctic clouds can reach as high as 1% if aerosol concentrations are low, further supporting the potential climatic importance of newly formed particles (Bulatovic et al., 2021)."*

Added references:
Bulatovic, I., Igel, A. L., Leck, C., Heintzenberg, J., Riipinen, I., and Ekman, A. M. L.: The importance of Aitken mode aerosol particles for cloud sustenance in the summertime high Arctic – a simulation study supported by observational data, 21, 3871–3897, https://doi.org/10.5194/acp-21-3871-2021, 2021.

Leaitch, W. R., Korolev, A., Aliabadi, A. A., Burkart, J., Willis, M. D., Abbatt, J. P. D., Bozem, H., Hoor, P., Köllner, F., Schneider, J., Herber, A., Konrad, C., and Brauner, R.: Effects of 20– 100 nm particles on liquid clouds in the cleansummertime Arctic, Atmos. Chem. Phys., 16, 11107–11124, https://doi.org/10.5194/acp-16-11107-2016, 2016.

P2L54-P3L56: about the possible influence of iodine contributing to the nucleation and growth, there is a new study from He et al. 2021 that could be mentioned maybe like this:

" There is evidence that iodine (e.g., iodine oxoacids) can also contribute to nucleation, usually on the coast of Greenland (Allan et al., 2015; Dall'Osto et al., 2018; Sipilä et al., 2016; He et al., 2021), as well as more recently in the Central Arctic Ocean (Baccarini et al., 2020)."

Additionally, Baccarini et al, 2020 observed that HIO3 was the main driver of growth to CCN sizes in central Arctic." However, there is typically insufficient iodine to explain the observed growth in these studies.". One location out of three-four locations is significant enough to be out of the "typically" word, maybe?

We thank the reviewer for pointing out this new publication as well as our unclear wording. The new reference has been added and the text has been edited to:

*"There is evidence that iodine (e.g. iodine oxoacids) can also contribute to nucleation and growth on the coast of Greenland (Allan et al., 2015; Dall'Osto et al., 2018; Sipilä et al., 2016), as well as more recently in the Central Arctic Ocean (Baccarini et al., 2020) and under controlled laboratory settings (He et al., 2021). However, the observed iodine is not always sufficient to explain the observed growth (Baccarini et al., 2020)."*

Added reference:
He, X.-C., Tham, Y. J., Dada, L., Wang, M., Finkenzeller, H., Stolzenburg, D., Iyer, S., Simon, M., Kürten, A., Shen, J., Rörup, B., Rissanen, M., Schobesberger, S., Baalbaki, R., Wang, D. S., Koenig, T. K., Jokinen, T., Sarnela, N., Beck, L. J., Almeida, J., Amanatidis, S., Amorim, A., Ataei, F., Baccarini, A., Bertozzi, B., Bianchi, F., Brilke, S., Caudillo, L., Chen, D., Chiu, R., Chu, B., Dias, A., Ding, A., Dommen, J., Duplissy, J., El Haddad, I., Gonzalez Carracedo, L., Granzin, M., Hansel, A., Heinritzi, M., Hofbauer, V., Junninen, H., Kangasluoma, J., Kemppainen, D., Kim, C., Kong, W., Krechmer, J. E., Kvashin, A., Laitinen, T., Lamkaddam,

H., Lee, C. P., Lehtipalo, K., Leiminger, M., Li, Z., Makhmutov, V., Manninen, H. E., Marie, G., Marten, R., Mathot, S., Mauldin, R. L., Mentler, B., Möhler, O., Müller, T., Nie, W., Onnela, A., Petäjä, T., Pfeifer, J., Philippov, M., Ranjithkumar, A., Saiz-Lopez, A., Salma, I., Scholz, W., Schuchmann, S., Schulze, B., Steiner, G., Stozhkov, Y., Tauber, C., Tomé, A., Thakur, R. C., Väisänen, O., Vazquez-Pufleau, M., Wagner, A. C., Wang, Y., Weber, S. K., Winkler, P. M., Wu, Y., Xiao, M., Yan, C., Ye, Q., Ylisirniö, A., Zauner-Wieczorek, M., Zha, Q., Zhou, P., Flagan, R. C., Curtius, J., Baltensperger, U., Kulmala, M., Kerminen, V.-M., Kurtén, T., et al.: Role of iodine oxoacids in atmospheric aerosol nucleation, 371, 589–595, https://doi.org/10.1126/science.abe0298, 2021.

P3-L59-60: There is for example a paper from Sihto et al 2011, looking specifically at CCN activation of these particles secondary organic aerosol from NPF and freshly grown (from the boreal forest). I m not sure how relevant is secondary organic aerosol from the boreal forest for the Canadian Arctic Archipelago comparison. However, the authors found similar results to this present paper.

We thank the reviewer for suggesting this paper. Reference to it has been made on lines 62 and 252-254 in the revised manuscript:

"*More recently, a growing body of work is providing evidence that organic compounds contribute to the growth of newly formed particle at lower latitudes (Ehn et al., 2014; Sihto et al., 2011)*"

"*Finally, the reported values are consistent with the κ value reported for pure SOA inferred from ambient observations at continental sites (e.g. Sihto et al., 2011; Gunthe et al., 2009).*"

Added references:
Gunthe, S. S., King, S. M., Rose, D., Chen, Q., Roldin, P., Farmer, D. K., Jimenez, J. L., Artaxo, P., Andreae, M. O., Martin, S. T., and Pöschl, U.: Cloud condensation nuclei in pristine tropical rainforest air of Amazonia: size-resolved measurements and modeling of atmospheric aerosol composition and CCN activity, 9, 7551–7575, https://doi.org/10.5194/acp-9-7551-2009, 2009.

Sihto, S.-L., Mikkilä, J., Vanhanen, J., Ehn, M., Liao, L., Lehtipalo, K., Aalto, P. P., Duplissy, J., Petäjä, T., Kerminen, V.-M., Boy, M., and Kulmala, M.: Seasonal variation of CCN concentrations and aerosol activation properties in boreal forest, Atmos. Chem. Phys., 11, 13269–13285, https://doi.org/10.5194/acp-11-13269-2011, 2011.

P4L105-110: I have three comments/questions there.

1) what was the sampling line? the flow is mentioned but not the length (i.e. would it be possible due to a very long sampling line and small sample flow 0.5 L min-1) that small particles are lost on the way to the CCN counter?

The inlet line from which the CCNC sampled was approximately 2 m long. However, other instruments also sampled from this line, with a total flow rate of 2.5 L min$^{-1}$, resulting in an estimated travel time of approximately 1.5 s before the particles reached the CCN counter. Using the particle loss calculator by von der Weiden et al. (2009), the estimated loss for 40 nm particles, the smallest sizes that were observed to activate, was approximately 2%. These details are now included in Sect. 2.1 as:

"*The CCNC was operated at a total flow rate of 0.5 L min$^{-1}$ and a sample flow rate of 0.05 L min$^{-1}$ with an estimated inlet residence time of 1.5 s and particle loss rate of <2% for the particles that activated in the chamber (>40 nm) (von der Weiden et al., 2009).*"

Added reference:
von der Weiden, S.-L., Drewnick, F., and Borrmann, S.: Particle Loss Calculator – a new software tool for the assessment of the performance of aerosol inlet systems, 2, 479–494, https://doi.org/10.5194/amt-2-479-2009, 2009.

2) Most important, what was the sampling temperature? Can the inlet flow slowly warm-up before entering the CCN counter? I'm wondering about possible evaporation of the aerosol in the sampling line, especially the organic volatile compounds of the aerosol sampled.

The reviewer brings up a very important point. The Scatterometer Lab, in which the CCNC was located, was unheated during the study and based on a limited number of temperature measurements throughout the cruise, varied between 4 and 20°C. In comparison, the ambient air temperature ranged from -2 to 16°C throughout the study period, suggesting that any warming in the inlet line before the CCNC would have been marginal. The following text has been added to Sect. 2.1 to address this point and the reviewer's concern is further addressed in the next point:

"*It is possible that the more volatile fractions of the ambient aerosol could have evaporated in the inlet line or in the CCNC. The room temperature was approximately 4-6 K warmer than the outside air temperature…*"

3) at which temperature the CCN counter was operating? How warmer compare to summer ambient Canadian cloud temperature? I always wonder (not only for this specific study), about the possible effect of the warm temperature in the CCN counter on how it can affect the partitioning of the volatile organic compounds of this artic SOA. I understand that it is a standard way to measure CCN properties worldwide and that this data set can be compared with other worldwide data, but maybe this could explain why all these organic aerosols, measured at a warm temperature, show a weak hygroscopicity.

As the reviewer points out, this is a systematic issue with these types of measurements using this instrument. The top of the chamber was 2-3°C warmer than the inlet temperature, which, when combined with the room temperature (as discussed in the previous point), would result in the sample air warming by almost 6-8°C. In addition, the maximum chamber temperature, near the bottom of the column, was up to 20°C warmer than the inlet temperature, and could have

potentially evaporated volatile SOA components, although a significant amount of aerosol liquid water would have been present by that point. The effect of this warming has been documented by Asa-Awuku et al. (2009), who observed the CCN-activity decreasing with the heating of SOA produced from the oxidation of beta-caryophyllene, suggesting that the volatile fractions were more hygroscopic. On the other hand, the most volatile fraction of ambient SOA have been observed to be less oxidized (Huffman et al., 2009), which we would expect to be less hygroscopic (Massoli et al., 2010; Lambe et al., 2011; Chang et al., 2010). More recent results of CCN measurements behind a thermal denuder have shown that the κ inferred for ambient organics at an urban site increased with heating of 20°C (Mei et al., 2021) suggesting that the volatile fraction is less hygroscopic. Overall, it seems that the effect of heating on aerosol hygroscopicity depends on the specific nature of the organic aerosol present and it is unclear whether the hygroscopicity in our study would have been over- or under-estimated. The following discussion is now included in Sect 2.1 starting on line 159 to address the previous two points:

*"It is possible that the more volatile fractions of the ambient aerosol could have evaporated in the inlet line or in the CCNC. The room temperature was approximately 4-6 K warmer than the outside air temperature, the top of the CCNC's chamber was approximately 2-3 K warmer than the inlet temperature, and the bottom of the chamber was up to 20 K warmer than the inlet temperature, although a significant amount of aerosol liquid water would have been present by that point. The effect of a 20 K warming has been observed to reduce the CCN-activity of laboratory-generated secondary organic aerosol (SOA), suggesting that the volatile fraction can be more hygroscopic (Asa-Awuku et al., 2009). On the other hand, the most volatile fraction of ambient SOA has been observed to be less oxidized (Huffman et al., 2009), which we would expect to be less hygroscopic (Massoli et al., 2010; Lambe et al., 2011; Chang et al., 2010). More recent results of CCN measurements behind a thermal denuder inferred that the κ value for ambient organics at an urban site increased with heating of 20 K (Mei et al., 2021) suggesting that the volatile fraction was less hygroscopic. Overall, it seems that the effect of heating on aerosol hygroscopicity depends on the specific nature of the organic aerosol present and it is unclear whether the hygroscopicity estimated in our study are potentially over- or under-estimated. As such, no corrections were applied for this warming, although it should be considered when interpreting our results."*

Added references:

Asa-Awuku, A., Engelhart, G. J., Lee, B. H., Pandis, S. N., and Nenes, A.: Relating CCN activity, volatility, and droplet growth kinetics of β-caryophyllene secondary organic aerosol, 9, 795–812, https://doi.org/10.5194/acp-9-795-2009, 2009.

Chang, R. Y.-W., Slowik, J. G., Shantz, N. C., Vlasenko, A., Liggio, J., Sjostedt, S. J., Leaitch, W. R., and Abbatt, J. P. D.: The hygroscopicity parameter (κ) of ambient organic aerosol at a field site subject to biogenic and anthropogenic influences: relationship to degree of aerosol oxidation, Atmos. Chem. Phys., 10, 5047–5064, https://doi.org/10.5194/acp-10-5047-2010, 2010.

Lambe, A. T., Onasch, T. B., Massoli, P., Croasdale, D. R., Wright, J. P., Ahern, A. T., Williams, L. R., Worsnop, D. R., Brune, W. H., and Davidovits, P.: Laboratory studies of the

chemical composition and cloud condensation nuclei (CCN) activity of secondary organic aerosol (SOA) and oxidized primary organic aerosol (OPOA), 11, 8913–8928, https://doi.org/10.5194/acp-11-8913-2011, 2011.

Massoli, P., Lambe, A. T., Ahern, A. T., Williams, L. R., Ehn, M., Mikkilä, J., Canagaratna, M. R., Brune, W. H., Onasch, T. B., Jayne, J. T., Petäjä, T., Kulmala, M., Laaksonen, A., Kolb, C. E., Davidovits, P., and Worsnop, D. R.: Relationship between aerosol oxidation level and hygroscopic properties of laboratory generated secondary organic aerosol (SOA) particles, 37, https://doi.org/10.1029/2010GL045258, 2010.

Mei, F., Wang, J., Zhou, S., Zhang, Q., Collier, S., and Xu, J.: Measurement report: Cloud condensation nuclei activity and its variation with organic oxidation level and volatility observed during an aerosol life cycle intensive operational period (ALC-IOP), 21, 13019–13029, https://doi.org/10.5194/acp-21-13019-2021, 2021.

*P6-L165: I m wondering about this median activation ratio absolute value. AR is defined as the ratio of the CCN concentration / NTot. While I could understand that CCN measured concentration can be seen as an absolute value (i.e. the smaller particle not reaching the CCN counter would not activate anyway), the Ntot correspond to the number of particles bigger than the detection limit of the specific DMPS (here 10 nm). So I m not sure how someone can use/compare these AR values with previous/future studies. I mean if instead, the DMPS measured from 15 nm onward for example, and the absolute value of AR would have jumped to 0.6 for example, would the authors have reached a different conclusion?*

The reviewer brings up a subtle but important point. Unfortunately, most observational studies (including those listed in Table 2) are limited by the most affordable condensation particle counter from TSI Inc. and typically only measure particles > 10 nm, the same as our study. As such, they should be comparable. To address the reviewer's concern here and the next point, the size range of particles represented by Ntot are now included in Table 2 and the text in this section has now been edited to read:

*"However, the median activation ratio (AR), defined as the CCN concentration / $N_T$, was only 0.38 at the highest SS of 0.99%, suggesting that most of the particles larger than 10 nm were either too small and/or non-hygroscopic to activate. These low AR are consistent with observations from Ny Alesund (Jung et al., 2018), where summer time AR of 0.4 for particles between 10 and 561 nm were reported, as well as other Arctic sites with a similar lower cut-off for $N_T$ (Table 2)."*

*P8-L167-168, the AR reported from the other studies, are they calculated with similar DMPS cut-off sizes (min and max)? The authors compare AR from Jung et al 2018 study, however, there is no value for Ntot reported in Table 2 for this specific study.*

An additional column has been added to Table 2 that includes the size range represented by $N_T$ from other studies. In addition, $N_T$ from the summer portion of the Jung et al. (2018) study has

also been included (195 cm$^{-3}$). Interestingly, all but one of the studies have a comparable size range as our study and would suggest that our comparisons are not completely meaningless.

Table 1: I m wondering if in Table 1 an extra value such as Dcrit should be explicitly reported. Dcrit can be calculated back using the reported kappa value (which has been calculated from Dcrit and with the assumption of the chemical composition of the artic aerosol sampled), but it is not as straightforward compared to having it directly reported. Dcrit can be used in certain regional/global modeling.

Thank you to the reviewer for suggesting this addition. Table 1 has now been modified to include the calculated Dcrit.

P9 L196-197: "Based on the median calculated $\kappa$ values from the entire study, we infer that >80% of the aerosol volume fraction was composed of a non-hygroscopic component, which we interpret as being organic". I m wondering if the word "non-hygroscopic" is too strong here (and in the manuscript in general). Multiple simulation chambers studies have shown, that SOA hygroscopicity, although not high, has a kappa bigger than 0, which is definitively more than "non-hygroscopic". For example, Bouzidi et al. 2022 (and reference therein) describe the hygroscopicity of SOA as "slightly hygroscopic" for their less hygroscopic case

Thank you to the reviewer for pointing out the poor choice of wording. We agree that non-hygroscopic is too strong and have changed all instances throughout the manuscript to "slightly hygroscopic". In addition, we now reference other ambient CCN measurements of SOA that have found similar values on line 250.

*"Finally, the reported values are consistent with the $\kappa$ value reported for pure SOA inferred from ambient observations at continental sites (e.g. Sihto et al., 2011; Gunthe et al., 2009)."*

Figure 1: I m not sure if it is possible to color the sea and the land to help to read the map? If possible, this would made the map easier to read.

This is an excellent suggestion. Figure 1 has now been edited so that the land is shaded light brown (see below). The water bodies remain white to avoid detracting from the blue datapoints but this will hopefully make it clearer for the reader.

[Figure]

Typos:

P2 L 23: "ref". Reference is missing.

The following references have been added at this line:
Brennan, M. K., Hakim, G. J., and Blanchard-Wrigglesworth, E.: Arctic Sea-Ice Variability During the Instrumental Era, 47, e2019GL086843, https://doi.org/10.1029/2019GL086843, 2020.

Stroeve, J. and Notz, D.: Changing state of Arctic sea ice across all seasons, Environ. Res. Lett., 13, 103001, https://doi.org/10.1088/1748-9326/aade56, 2018.

This study analyzes the ultrafine particles (UFP) and/or growth and their potential contribution to cloud properties in the eastern Canadian Arctic Archipelago. The particle size distribution and cloud condensation nuclei (CCN) concentration were measured on board the CCGS Amundsen during summertime in 2016 when UPF and growth events were frequently occurred. Low hygroscopicity parameter ($\kappa$) implies that significant proportion of particles were non-hygroscopic, like organic, mostly originated from open water within the Arctic and the $\kappa$ value is comparable with other studies. The CCN concentrations increased during all UFP and/or growth events suggest that small particles in Arctic region can affect cloud radiative properties by contributing to cloud droplet number concentration (CDNC). The paper is well organized with data description, analysis and conclusions that have meaningful implications to the Arctic study. The manuscript is relevant to the readers of ACP and I recommend publication if the following comments can be addressed.

Line 23: Add reference ('summer sea ice extent that is steadily declining [ref].')

The following references have been added at this line:
Brennan, M. K., Hakim, G. J., and Blanchard-Wrigglesworth, E.: Arctic Sea-Ice Variability During the Instrumental Era, 47, e2019GL086843, https://doi.org/10.1029/2019GL086843, 2020.

Stroeve, J. and Notz, D.: Changing state of Arctic sea ice across all seasons, Environ. Res. Lett., 13, 103001, https://doi.org/10.1088/1748-9326/aade56, 2018.

Line 140: The method for deriving $D_a$ contains an important assumption that particles are internally mixed. Is an internal mixture assumption of particle reasonable in this region? What about externally mixed particles?

The reviewer raises a very important point. An externally-mixed aerosol population is always a possibility and a concern in these types of calculations. However, this likely does not affect our calculations of Da significantly because of several reasons. Firstly, the summer Arctic aerosol typically originates from local sources, reducing the likelihood of a separate mode with different properties transported from afar. Secondly, although primary emissions from Arctic sources, such as sea spray or dust, could potentially contribute to the aerosol being externally-mixed, these particles would be expected to be present at larger sizes and not affect the calculation of Da. Finally, even if the smaller particles that affect our calculations were externally-mixed, the frequently-observed process of condensation leading to growth would cause the aerosol to become more similar in composition, thus reducing the errors caused by an externally-mixed population. To address the reviewer's concern, this following text has been added to this section starting on line 152:

*"Critical assumptions in this approach are that the particles are uniform in composition at a given size and that all the particles larger than Da activate. These assumptions are unlikely to significantly affect our calculations because the summer Arctic aerosol typically originates from local sources, reducing the likelihood of a separate mode with different properties transported*

*from afar. In addition, although primary emissions from Arctic sources, such as sea spray or dust, could potentially contribute to the aerosol being externally-mixed, these particles would be expected to be present at larger sizes and not affect the calculation of Da."*

As well as to Section 3.1 (Line 200):

*"In addition, any externally-mixed aerosol population that existed before these growth events would become more similar in composition due to condensational growth, thus reducing errors associated with the assumptions used to calculate κ using Eq. 1."*

Line 150: The author defines the background period as times when $N_T < 500$ cm$^{-3}$? Is it based on the measurement (average value except for UFP and growth event? Or total average value?) or based on the reference? Add an explanation in the Section 2.4.

The limit of 500 cm$^{-3}$ is rounded up from the median concentration during the study (472 cm$^{-3}$). As seen in the figure below, the probability distribution is heavily skewed to the lower concentrations and has a long tail at the higher concentrations. It is also close to the 95$^{th}$ percentile (484 cm$^{-3}$) of the concentrations when $D_{max} > 100$ nm (i.e. when accumulation mode particles dominate the distribution). The following sentence has been added to Sect. 2.4

*"The concentration limit of 500 cm$^{-3}$ was determined by considering various statistical values including the median $N_T$ over the entire study (472 cm$^{-3}$) and the 95$^{th}$ percentile of $N_T$ when $D_{max} > 100$ nm (484 cm$^{-3}$)."*

[Figure]

Figure 1 – Histogram of SMPS concentrations during the study. Vertical red line indicates 500 cm$^{-3}$.

Line 157. For three regions, particle and CCN number concentration appear to be quite different as well as κ value. Is this difference by regional effect? Or air mass back trajectory effect during each period? it would be better to add detailed description of Figure 2 and Table 1.

Attributing the differences in aerosol distributions and hygroscopicity in the different legs is the biggest challenge of this study since not only was the geographical region changing, but also the seasonal effects on biology as well as air mass trajectory. The 5-day air mass history calculated from FLEXPART showed that the influence from the lowest 200 m was primarily confined to the immediate region of the ship, suggesting that regional effects were important (for example, see Figures 5e-5h from the study by Collins et al. (2017) included below). However, it is important to note that it was likely the local conditions in each region (e.g. sea-ice coverage, solar radiation, biological activity, cloud cover, wind speed), that actually contributed to the variations in the aerosol population. This is discussed by Collins et al. (2017), who found that although ultrafine particles were observed extensively in our study region in the summer of 2016, they were almost completely absent during the 2014 NETCARE cruise which occurred at a similar time of year, suggesting that geography alone was insufficient to explain our events. The three legs in our study were conveniently defined by geographical region and could only capture some of the differences in the environmental conditions experienced during our study. To address the reviewer's concerns, we have added additional text describing the results from the three legs presented in Figure 2 and Table 1.

[Figure]

Figure 5 from Collins et al. (2017) – FLEXPART air mass histories extending backwards 5 days from the time noted at the top of each pair of plots. The top rows (a–d) represent time-integrated PES for a 0–10 km column, and the bottom rows (e–h) represent time-integrated PES for a 0–200m column. Note the difference in projection and spatial scale in the panels (e–h) to emphasize the local detail of the 0–200m PES. Each of the release times corresponds to the observation of a UFP formation and/or growth event aboard CCGS Amundsen. Tracer release locations were dictated by the ship's coordinates at the given release time. Details on the model are described in Sect. 2.2 of Collins et al. (2017).

In Sect. 2.1, the following text was edited:

"*the study was conveniently divided into three legs based on geography*"

the following text was removed:

"*Furthermore, the latitudinal change was greatest in the Nares strait region, while the other two regions had greater variations in longitude.*"

and replaced with:

*Collins et al. (2017) showed that the 5-day backward simulations of surface (< 200 m) influence using the FLEXible PARTicle dispersion model (FLEXPART) was primarily confined to the immediate region of the ship, suggesting that regional emissions were important during our study. However, it is difficult to separate the effects of geography and season on the observed differences in the aerosol population since the local conditions in each region (e.g. sea-ice coverage, solar radiation, biological activity, cloud cover, wind speed) would have affected the observed aerosols.*

In the first paragraph of Section 3.1 (line 189), the following text was added:

*"These events were especially prevalent in the first part of the study when the ship was in the warmer and more biologically-active waters of Baffin Bay and persisted into the southern portion of Nares Strait into Kane Basin of the 2$^{nd}$ leg. The most northern extent of the cruise occurred on 13 August, where the ship encountered sea-ice and very few UFP were observed. Only one UFP and/or growth event was observed during the last leg of the study where the waters were ice-free, shallower and less saline. It is possible that the local conditions were less favourable for the formation of UFP, although it should be noted that this period was much shorter than the other legs (six days compared to 12–13 days)."*

And the text describing the regional effects of kappa on lines 225-232 of the revised manuscript were edited to reflect this change and to address the reviewer's comments:

*"The values also increased throughout the study, with the lowest values (0.04–0.08) corresponding to the first part of the study when the ship was in the more southerly, warmer and open waters of Baffin Bay where more UFP and growth events were observed, and the highest values (0.18–0.21) corresponding to the shallower and more coastal waters during Leg 3 when only one UFP and growth event occurred. Based on our observations, it is unclear whether the increase in hygroscopicity driven by a reduction in UFP and growth events was caused by changes in emissions in the different regions such as more land influence in the Resolute leg, or whether it is due to the fact that we sampled in this region two weeks later when the biological activity had changed. It should be noted that the 2014 NETCARE cruise in the Queen Maud Gulf region west of Resolute Bay did not observe any UFP events in early August (Collins et al., 2017) while Chang et al. (2011) reported one UFP and growth event in late August in 2008 in the same region. Overall, this suggests that local environmental conditions are the main driver of aerosol sources and therefore hygroscopicity."*

In connection with the previous comment, can the author expect that all three regions give a same conclusion when the plot the same graph with Fig.5 by dividing it into three regions, not the whole regions?

The reviewer brings up an interesting point. Figure 2 below shows the change in CCN concentrations for all UFP and/or Growth times (10 nm < Dmax < 100 nm and $N_T$ > 500 cm$^{-3}$) for the three legs (red, blue and green, respectively) and the whole study (thick black line). Figure 3 shows only times when Dmax > 40 nm. In both cases, Legs 1 and 2 are very similar to the mean over the whole study. This is not surprising as during Leg 3, only two periods of UFP (consisting of 15-20 data points) were observed and no "Growth > 40 nm" times at all, which is why the total study mean does not reflect the observations from Leg 3. It would appear that UFP and/or Growth affect CCN concentrations to a similar degree during Legs 1 and 2 but not during Leg 3, likely due to the lack of UFP and/or Growth during Leg 3. The discussion in this section has now been updated to:

*"Because only one UFP event was observed in the last leg of the study when the ship was in the Lancaster Sound and Queen Maud Gulf, these results are only representative of the Baffin Bay and Nares Strait periods of the study."*

[Figure]

 Figure 2 – Increase in CCN concentrations for UFP and/or Growth times for the entire study (thick black line) as well as broken down by the three legs of the study (red, blue and green). The whiskers show the 95% confidence interval as described in the main text.

[Figure]

Figure 3 – Increase in CCN concentrations for Growth > 40 nm times for the entire study (thick black line) as well as broken down by the first two legs of the study (red and blue). No data for Leg 3 are shown because there were no Growth > 40 nm periods during that leg. The whiskers show the 95% confidence interval as described in the main text.

Line 271: This is not a mandatory but it might be clear if the author shows the history of air mass during the campaign period.

We did not include these in the original manuscript because similar results were previously included in Collins et al. (2017) (see Fig. 5 of their paper included above). The 1st paragraph in Section 3.1 now includes the following text:

*Collins et al. (2017) showed that the 5-day backward simulations of surface (< 200 m) influence using the FLEXible PARTicle dispersion model (FLEXPART) was primarily confined to the immediate region of the ship, suggesting that regional emissions were important during our study.*

Typographical errors

The following typographical errors were corrected in the revised manuscript:

Line 125: "*uses*" changed to "*used*"

Line 234: "*one event*" changed to "*one NPF event*"

Table 2: The "*Sub-Arctic*" descriptor was changed to "*Arctic*" to reflect the fact that the study by Herenz et al. (2018) was conducted at 69°N.